# Are skip connections necessary for biologically plausible learning rules?

**Daniel Jiwoong Im, Rutuja Patil, Kristin Branson**
Janelia Research Campus, HHMI
{imd,patilr,bransonk}@janelia.hhmi.org

## Abstract

Backpropagation is the workhorse of deep learning, however several other biologically-motivated learning rules have been introduced, such as random feedback alignment and difference target propagation. None of these methods have produced competitive performance against backpropagation. In this paper, we show that biologically-motivated learning rules with skip connections between intermediate layers can perform as well as backpropagation on the MNIST dataset and are robust to various sets of hyper-parameters.

## 1 Introduction

The backpropagation (BP) of global error [11] has been tremendously successful in solving hard AI tasks using deep learning. All deep learning models, such as deep feed forward neural networks, recurrent neural networks, and deep reinforcement learning, use BP as the main credit assignment tool to update their weights (model parameters) [6]. However, BP for the brain is widely considered to be biologically implausible because BP requires symmetric backward connectivity patterns (weight transpose) and it does not deliver the error signals through a distinct pathway. Such concerns have inspired researchers to develop biologically-motivated learning rules[1] while trying to attain the performance of artificial neural networks. The two most popular learning rules are random feedback alignment (FA) [8] and difference target propagation (DTP) [7]. Instead of having symmetric weight connections, FA uses random feedback weights as a backward pathway to propagate the error information. DTP trains separate feedback neural network that produces the target activities and then minimizes the error between target activities and forward propagated activities. Several other methods were inspired by FA and DTP, such as directed feedback alignment [9] and simplified difference target propagation (SDTP) [1]. However, *Bartunov et. al.* demonstrate that not only are these methods non scalable to problems like the ImageNet dataset, but also that performance decays as more biologically plausible constraints are added [1]. We show in addition that performance is not robust with respect to hyper-parameters for variants of DTP methods.

The concept of skip and dense connections in deep learning was first introduced from residual neural networks[4] and densely connected convolutional networks [5], wherein skip implies later layers receive signal from the earlier layers with intermediate skips, whereas dense implies each layer is connected to each other. The brain contains of many examples of both skip and dense connections. For example, the neocortex has a similar structure to residual nets; where cortical layer VI neurons get input from layer I, which skips intermediate layers [12]. Similar skip connection structure exist in multiple other layers [3]. *Oh et. al.* [10] laid out the adult mouse brain mesoscale connectome and showed skip connections between inter-regions. The whole-brain and corticocortical connections can be fit by one-component lognormal distributions. In general, the log-normal distribution connectivity implies that sparse long-range connections exist in the brain, which may act as skip connections in our context [2, 10].

---

[1]We refer to credit assignment methods as learning rules.

| | Sigmoid | | | Relu | | |
|---|---|---|---|---|---|---|
| | Learning Rate | Early Stop | Depth | Learning Rate | Early Stop | Depth |
| BP | (0.1-0.001) | (200k-800k) | (3-7) | (0.001-0.00001) | (100k-300k) | (5-10) |
| FA | (0.01-0.0001) | (200k-800k) | (3-7) | (0.001-0.00001) | (600k-1000k) | (5-10) |
| DTP | (0.01-0.0001) | (200k-800k) | (3-7) | (0.001-0.00001) | (50k-150k)[2] | (5-10) |

Table 1: The table provides a range of hyper-parameters explored for various learning rules. Throughout the hyper-parameter search, we explored three sets of learning rates and five sets of the early stopping starting points. We tried 0.001, 1e-3, and 1e-4 learning rates and 0.01, 0.001, and 1e-03 learning rates for the architectures with ReLU and sigmoid activations respectively. We tried 200k,400k,600k,800k, 1,000k early stop starting point for NN and DN. We tried three different range of early stop starting points that are uniformly spaced out for ConvNet and DenseConvNet.

The performance of computer vision and natural language processing methods has improved over the last five years by increasing the depth of deep neural networks. With the introduction of skip and dense connections from residual neural networks [4] and densely connected convolutional networks [5], training with hundreds even thousands of layers has became possible. The core idea behind the performance gain with skip and dense connections is that a shorter path from earlier layers to later layers is created, and information as gradients gets propagated more efficiently through more layers. In our experiments, we demonstrate that such type of skip connections help even more for biologically motivated learning rules.

Taking inspiration from the connectivity in the brain and taking the performance advancement in deep skip and dense networks as exemplars, we show that skip and dense connections allow biologically-plausible learning rules to perform as well as backpropagation. We show that FA and DTP with densely connected deep neural networks perform comparable to BP even with an increase in depth, and show that they are much more robust against different hyper-parameters compared to non-densely connected networks.

## 2   Methods

We use fully connected neural network and convolutional network architectures with dense connectivity. Dense connectivity refers to direct connections from any layer to all subsequent layers. More formally, we define densely connected neural network to be $\mathbf{h}_l = f_l([h_1; \cdots ; h_{l-1}]; \theta_l) = \sigma(W_{l,1}h_1 + \cdots W_{l,l-1}h_{l-1} + b_l)$, where $\theta_l = \{W_{l,1}, \cdots , W_{l,l-1}, b_l\}$ are the parameters of neural network with weights $W_{l,l-1}$ connecting from layer $l-1$ to $l$, and $h_0 = x$. $[\cdot; \cdot]$ refers to the concatenation between vectors.

We can use standard BP on dense network to learn the weights. The gradient of hidden layer $l$ and parameter $\theta_l$ can be derived using chain rule: $\frac{d\mathcal{L}}{dh_i} = \sum_{i=1}^{L} \left(\frac{dh_i}{dh_l}\right)^T \frac{d\mathcal{L}}{dh_i}$ and $\frac{d\mathcal{L}}{d\theta_l} = \left(\frac{dh_l}{d\theta_l}\right)^T \frac{d\mathcal{L}}{dh_l}$. Similarly for FA, we can replace the transpose weight matrices with fixed random connections. For DTP, the decoder network is defined as $\hat{h}_l = g(h_{l+1}; \lambda_{l+1})$ which is learned to act as an inverse transformation $f^{-1}(h_{l+1}; \theta_{l+1})$. Then, the target activation $l$, $\hat{h}_l$, becomes $h_l \leftarrow h_l - \sum_{j=l+1}^{L}(g_j(h_j) - g_j(\hat{h}_j))$. We can minimize the standard difference target loss and reconstruction loss for $\theta$ and $\lambda$ [7].

Note that all the above can be easily extended to convolutional neural network.

## 3   Experiment

We conducted our experiments on different learning methods with different network architectures on the MNIST dataset. We compared the performance of feedfoward neural network (NN) against dense neural network (DN) and convolution network (ConvNet) against dense convolutional network (DenseConvNet) for BP, FA and DTP. It is well known that BP without batch normalization suffers from vanishing gradients as the feedfoward neural network depth increases, especially with sigmoid

---

[2]Early stopping range between 200k-600k explored for multi-lyer perceptron (NN and DN) and 50k-150k explored for convolutional neural networks (ConNet and DenseConvNet)

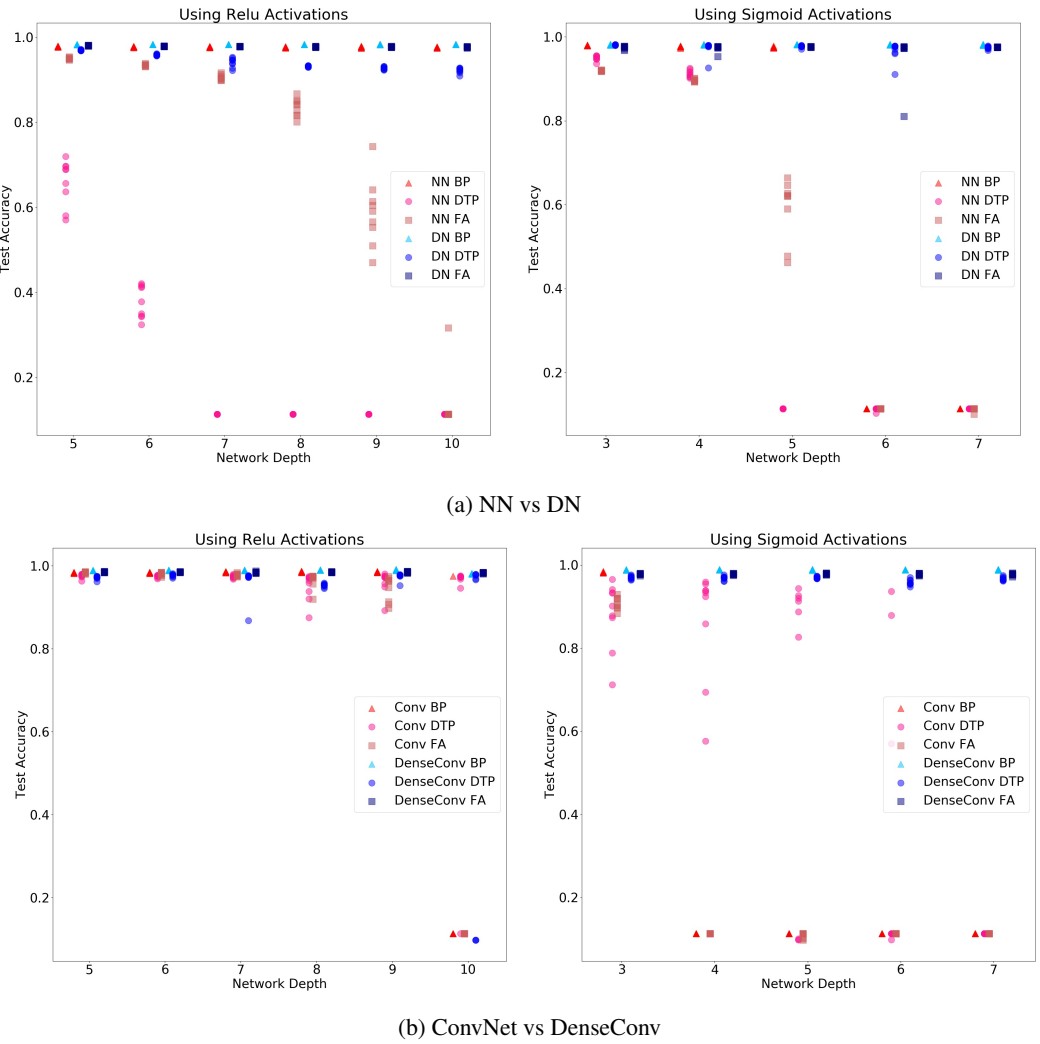

(a) NN vs DN

(b) ConvNet vs DenseConv

Figure 1: The performance over BP, FA, DTP with respect to different network depths.

activation function. The same holds for FA and DTP as well. We want to evaluate the performance of FA and DTP as we increase network depth for densely connected networks. Furthermore, we want to know whether they are more robust to different sets of hyper-parameters.

In our experiments, the dataset was divided into 50,000 training, 10,000 validation, and 10,000 test. We trained each network with a batch size of 128 and a 0.00001 L2 weight decay coefficient. Table 1 presents the set of hyper-parameters we tested. We explored learning rates between 0.00001 and 0.1 and explored early stopping criterion starting points between 20,000 and 1,000,000. We used 128 hidden units for each fully connected hidden layer. We used a convolutional filter size of three and channel size (depth x 16) for convolutional neural networks. We explored both sigmoid and ReLu activation functions for multi-layer perceptron and convolutional neural networks. We measured the performance of the model with network depth from three to seven layers for sigmoid activation and five to ten layers for ReLu activation.

Figure 1 presents the test accuracy over BP, FA, and DTP with respect to network depth. The results of BP, FA, DTP are paired with NN and DN in Figure 1a, and paired with ConvNet, and DenseConvNet in Figure 1b. The best hyper-parameters for each model is chosen across 10 folds. We observe that test accuracy for all three methods drop for NN and ConvNet with network depths, whereas the test accuracy maintains for DN and DenseConvNet. This illustrates that the network did not suffer from propagating error signals all the way to bottom layers when having dense connections. It is well-known that BP suffers from vanishing gradients with deep neural networks, and yet the

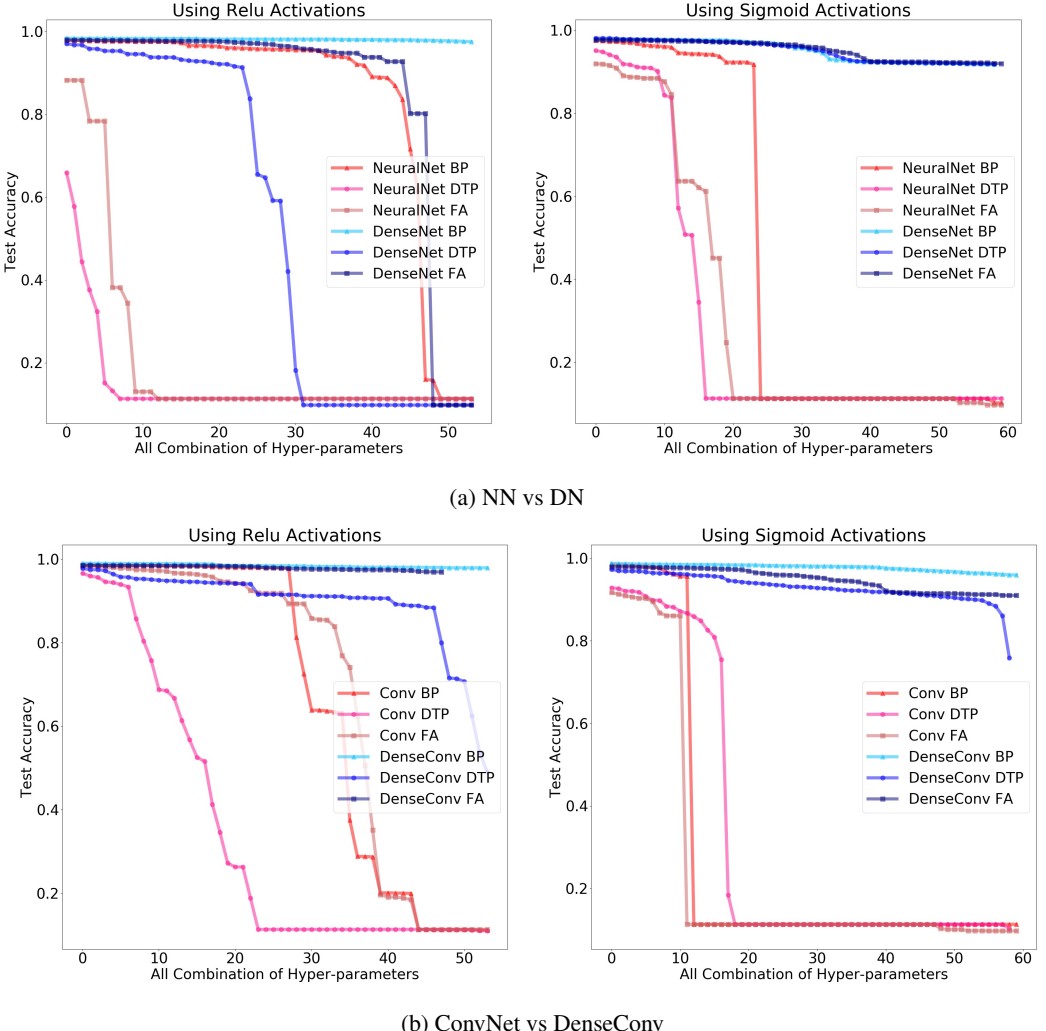

(a) NN vs DN

(b) ConvNet vs DenseConv

Figure 2: The sensitivity of test accuracy over multiple hyper-parameter for BP, FA, DTP. X-axis are all combination of hyper-parameter settings for learning rates, epochs, and network depth (sorted based on accuracy rate).

dense connectivity allows short pathway from the top layers to bottom layers. We suspect that FA benefits from the dense connectivity the same way as BP. Even though DTP is a local learning rule, the activities in the top layers need to be informative in order for the local weight updates to make sense, otherwise, the local updates are based on random signals from the adjacent layers for the fully connected neural network. However, the dense connectivity will enable transfer of information from the top layers to bottom, since every hidden layer are adjacent layers from each other.

Figure 2 presents the sensitivity of test accuracy over multiple hyper-parameters. We explored combinations of learning rates, epochs, and depth, which are ordered based on sorted test accuracy. We can see that the performance of FA and DTP for NN and ConvNet (red lines) varies across a wide range. In fact, there are big accuracy discrepancy for depth 3, 4, and 5 for NN and ConvNet. However, the performance of all three learning rules for DN and DenseConvNet (blue lines) remain nearly constant. This illustrates that having dense connections makes the model more robust to different hyper-parameters.

## 4 Discussion

Building an intelligent system may require an appropriate objective/reward function, credit assignment method, specific architecture types, or some combination of all of the above. The nervous systems of animals illustrate some of the building blocks necessary for building intelligent systems, such as dopamine-based reward signals, distinct error signal pathways and neuronal architectures capable of performing computations. However, the error rates for the ImageNet classification challenge using AlexNet are in the range of 93~98% for all biologically motivated learning rules, whereas the BP error rate is 63.9% [1]. BP is therefore more effective than biologically motivated learning rules. However, BP cannot be employed in biologically plausible architectures because it requires symmetric backward connectivity and does not have a distinct error signal propagation pathway. Why are biologically-plausible learning rules so ineffective in the context of deep learning? We believe it is because biologically-inspired learning rules have been studied in isolation, rather than considering them in the context of biologically-constrained architectures. Thus, re-examining the other key biological conditions that induce better learning performance is required. In this paper, we posit that the skip connections in nervous systems could be one of the key architectural components that are required to enhance existing and still unexplored credit assignment methods. Through this experiment, we show that biologically-motivated learning rules like FA and DTP are more effective when combined with dense and skip connections. Furthermore, it is possible that having lognormal-distributed skip connections, as observed in the mouse brain, could be the computationally efficient way to propagate information. We leave this to future work.

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
