# OpenReview forum: "Are skip connections necessary for biologically plausible learning rules?"
_NeurIPS.cc/2019/Workshop/Neuro_AI — Real Neurons & Hidden Units @ NeurIPS 2019 Poster_

### Official Review · AnonReviewer3 · 2019-09-24
**Biologically plausible backprop is more plausible with skip connections**

**Clarity:** 3

**Comment:**

The question raised in this paper is very novel and opens up new directions for future research.
More clarity in the testing process can improve the conclusions: the choice of models, hyper-parameters, etc.
Also, adding the skip connections can potentially reduce the rate of change in the weights of hidden units that are not monosynaptically connected to the output layer. In a dense net, improvements in performance can be achieved by mainly tuning the weights that are one synapse away from the output layer, circumventing the deep credit assignment problem (the source of vanishing gradient problem). Showing the dynamics of weight changes across layers throughput learning can help understanding the way skip connections facilitate biological backprop.


**Category:**

AI->Neuro

**Clarity Comment:**

The motivation behind the study is clear. The results are presented in an understandable format. The only part that would benefit from more clarity is Figure 2 and the results related to this figure. The horizontal axis in Figure 2 is labeled with "all combination of hyper-paramteres", which is not quite clear what it's supposed to mean. From the models' performance, it seems that by going from 0 to 60 along the horizontal axis, the complexity of the models is increasing, though it's not clear at all what these numbers correspond to.

**Evaluation:**

3: Good

**Importance:**

3: Important

**Importance Comment:**

The biological plausibility of deep learning models is an important topic of research both in neuroscience and AI. Recent studies have emphasized the importance of model architecture for having a more biologically realistic representations across different layers of deep nets (e.g. recent studies from DiCarlo's lab). In this paper, it's argued that model architecture might also matter for the success of biologically plausible versions of back-propagation such as feedback alignment (FA).


**Intersection:**

4: High

**Intersection Comment:**

This paper is related to important open questions in both neuroscience and AI.
Skip connection, as they were proposed in the deep learning literature, could solve important learning problems such as vanishing gradients. The functional importance of skip connections has also gained some attention recently in neuroscience. This paper lies at the intersection of the both fields: the computational efficiency that skip connections offer to deep learning turned out to be important for biologically plausible BP as well.

**Rigor Comment:**

The authors have compared the accuracy of the trained models on a test set for different hyperparameter values. The reported results support their main claim that after including skip connections in the model architecture, FA and DTP (two biologically plausible versions of BP) are as successful in training the model as the ordinary BP. It has been known that adding skip connections partially solves the problem of vanishing gradients with BP. It was conjectured that algorithms such as FA and DTP are not as successful in managing vanishing gradients (especially in very deep architectures) due to their less rigorous error back-propagation. This study provides yet another evidence for the importance of skip connections in dealing with vanishing gradients; a feature that also renders algorithms such as FA and DTP more successful in practice.
The results presented in Figure 2 could benefit from more clarity. The hyper-parameters that were included in the test, and the way they were changed throughout different tests are not very clear.

**Technical Rigor:**

3: Convincing

---

### Official Review · AnonReviewer2 · 2019-09-26
**Skip connections combined with biologically motivated learning rules**

**Clarity:** 3

**Comment:**

It would be great if authors also use ResNet architecture and compare their results with that. Also, providing results for larger datasets such as CIFAR-10 or ImageNet with deeper networks would be useful.


**Category:**

AI->Neuro

**Clarity Comment:**

In figure 2, it is not clear what numbers for “all combinations of hyper-parameters” exactly correspond to and how hyper-parameters like depth, learning rate, and the number of epochs were changed in different models tested.


**Evaluation:**

3: Good

**Importance:**

3: Important

**Importance Comment:**

There has been a growing disagreement about whether the backpropagation (BP) can explain learning in the brain. Many biologically plausible algorithms have been proposed as alternatives for BP. Although these algorithms move toward biological plausibility, biologically inspired model architectures also need to be taken into account. This work looks at a biologically plausible model architecture in neural networks and argues that skip connections can improve the performance of such algorithms.


**Intersection:**

4: High

**Intersection Comment:**

This paper is motivated by the questions that help to better understand the brain and look at current successful models for deep learning from a biological perspective. The idea of using skip connections with biologically plausible algorithms lies at the intersection of neuroscience and AI.


**Rigor Comment:**

Alternative BP algorithms such as FA and TP failed to get the same success of BP in deep networks. The authors claim that by using skip connections in the architecture they get the same results as with BP. Their experimental results support this claim. However, results for sensitivity of test accuracy for multiple hyper-parameters needs more clarification.


**Technical Rigor:**

3: Convincing

---

### Official Review · AnonReviewer1 · 2019-09-26
**Brief review of skip connections for biologically plausible learning rules**

**Clarity:** 4

**Comment:**

This is a very interesting set of results and are of interest broadly. Many of the ideas presented here could be followed up in many directions, and this architecture is interesting for the computation of biological connections.
Some explanation of the accuracy variations for depths 3,4, and 5 would be useful.

**Category:**

Neuro->AI

**Clarity Comment:**

Overall the work is well-written and easy to follow. The figures would benefit from some additional explanations of what the parameter spaces mean (as can be interpreted by someone who is less of an expert in their field), and also how the biological framework for skip connections was derived.

**Evaluation:**

5: Excellent

**Importance:**

4: Very important

**Importance Comment:**

This work suggests that backpropagation may benefit from more biologically inspired architectures. Recent developments in AI support this line of argument, and the authors further extend this by incorporating biological structure into these frameworks. Work such as from Svoboda's lab highlights that sparse long-range neural connections can have important functionality in the behaving brain, so that this improves performance in artificial networks is highly interesting.

**Intersection:**

4: High

**Intersection Comment:**

This work is very well situated between AI and neuroscience - as we are only beginning to fully understand the connectomic architecture of cortical layers in neuroscience, this may be an incredibly rich landscape in the future as we generate connectomic maps. The application of these connectivity structures to AI is a highly intersectional area, and future efforts in this area can inform both AI and neuroscience.

**Rigor Comment:**

The authors present data that skip connections can achieve the same performance as backpropagation in a neural network. This data is convincing, but may benefit from an increased parameter space and more detailed about the generation of skip connections.

**Technical Rigor:**

3: Convincing

---

### Decision · Program_Chairs · 2019-10-02

Accept (Poster)